# ANTISYMMETRICRNN: A DYNAMICAL SYSTEM VIEW ON RECURRENT NEURAL NETWORKS

**Bo Chang**[*]
University of British Columbia
Vancouver, BC, Canada
bchang@stat.ubc.ca

**Minmin Chen**
Google Brain
Mountain View, CA, USA
minminc@google.com

**Eldad Haber**
University of British Columbia
Vancouver, BC, Canada
haber@math.ubc.ca

**Ed H. Chi**
Google Brain
Mountain View, CA, USA
edchi@google.com

## ABSTRACT

Recurrent neural networks have gained widespread use in modeling sequential data. Learning long-term dependencies using these models remains difficult though, due to exploding or vanishing gradients. In this paper, we draw connections between recurrent networks and ordinary differential equations. A special form of recurrent networks called the AntisymmetricRNN is proposed under this theoretical framework, which is able to capture long-term dependencies thanks to the stability property of its underlying differential equation. Existing approaches to improving RNN trainability often incur significant computation overhead. In comparison, AntisymmetricRNN achieves the same goal by design. We showcase the advantage of this new architecture through extensive simulations and experiments. AntisymmetricRNN exhibits much more predictable dynamics. It outperforms regular LSTM models on tasks requiring long-term memory and matches the performance on tasks where short-term dependencies dominate despite being much simpler.

## 1 INTRODUCTION

Recurrent neural networks (RNNs) (Rumelhart et al., 1986; Elman, 1990) have found widespread use across a variety of domains from language modeling (Mikolov et al., 2010; Kiros et al., 2015; Jozefowicz et al., 2016) and machine translation (Bahdanau et al., 2014) to speech recognition (Graves et al., 2013) and recommendation systems (Hidasi et al., 2015; Wu et al., 2017). Modeling complex temporal dependencies in sequential data using RNNs, especially the long-term dependencies, remains an open challenge. The main difficulty arises as the error signal back-propagated through time (BPTT) suffers from exponential growth or decay, a dilemma commonly referred to as exploding or vanishing gradient (Pascanu et al., 2012; Bengio et al., 1994).

Vanilla RNNs as originally proposed are particularly prone to these issues and are rarely used in practice. Gated variants of RNNs, such as long short-term memory (LSTM) networks (Hochreiter & Schmidhuber, 1997) and gated recurrent units (GRU) (Cho et al., 2014) that feature various forms of "gating" are proposed to alleviate these issues. The gates allow information to flow from inputs at any previous time steps to the end of the sequence more easily, partially addressing the vanishing gradient problem (Collins et al., 2016). In practice, these models must be paired with techniques such as normalization layers (Ioffe & Szegedy, 2015; Ba et al., 2016) and gradient clipping (Pascanu et al., 2013) to achieve good performance.

Identity and orthogonal initialization is another proposed solution to the exploding or vanishing gradient problem of deep neural networks (Le et al., 2015; Mishkin & Matas, 2015; Saxe et al., 2013; Chen et al., 2018a). Recently, Arjovsky et al. (2016); Wisdom et al. (2016); Hyland & Rätsch

---

[*]Work performed while interning at Google Brain.

(2017); Xie et al. (2017); Zhang et al. (2018a) advocate going beyond initialization and forcing the weight matrices to be orthogonal throughout the entire learning process. However, some of these approaches come with significant computational overhead and reportedly hinder representation power of these models (Vorontsov et al., 2017). Moreover, orthogonal weight matrices alone do not prevent exploding and vanishing gradients, due to the nonlinear nature of deep neural networks as shown in (Pennington et al., 2017).

Here we offer a new perspective on the trainability of RNNs from the dynamical system viewpoint. While exploding gradient is a manifestation of the instability of the underlying dynamical system, vanishing gradient results from a lossy system, properties that have been widely studied in the dynamical system literature (Haber & Ruthotto, 2017; Laurent & von Brecht, 2017). The main contributions of the work are:

- We draw connections between RNNs and the ordinary differential equation theory and design new recurrent architectures by discretizing ODEs.

- The stability of the ODE solutions and the numerical methods for solving ODEs lead us to design a special form of RNNs, which we name *AntisymmetricRNN*, that can capture long-term dependencies in the inputs. The construction of the model is much simpler compared to the existing methods for improving RNN trainability.

- We conduct extensive simulations and experiments to demonstrate the benefits of this new RNN architecture. AntisymmetricRNN exhibits well-behaved dynamics and outperforms the regular LSTM model on tasks requiring long-term memory, and matches its performance on tasks where short-term dependencies dominate with much fewer parameters.

## 2 RELATED WORK

**Trainability of RNNs.** Capturing long-term dependencies using RNNs has been a long-standing research topic with various approaches proposed. The first group of approaches mitigates the exploding or vanishing gradient issues by introducing some forms of gating. Long short-term memory networks (LSTM) (Hochreiter & Schmidhuber, 1997) and gated recurrent units (GRU) (Cho et al., 2014) are the most prominent models along this line of work. Parallel efforts include designing special neural architectures, such as hierarchical RNNs (El Hihi & Bengio, 1996), recursive neural networks (Socher et al., 2011), attention networks (Bahdanau et al., 2014), dilated convolutions (Yu & Koltun, 2015), recurrent batch normalization (Cooijmans et al., 2017), residual RNNs (Yue et al., 2018), and Fourier recurrent units (Zhang et al., 2018b). These, however, are fundamentally different networks with different tradeoffs.

Another direction is to constrain the weight matrices of an RNN so that the back-propagated error signal is well conditioned, in particular, the input-output Jacobian having unitary singular values. Le et al. (2015); Mikolov et al. (2014); Mishkin & Matas (2015) propose the use of identity or orthogonal matrix to initialize the recurrent weight matrix. Followup works further constrain the weight matrix throughout the entire learning process either through re-parametrization (Arjovsky et al., 2016), geodesic gradient descent on the Stiefel manifold (Wisdom et al., 2016; Vorontsov et al., 2017), or constraining the singular values (Jose et al., 2017; Kanai et al., 2017; Zhang et al., 2018a). It is worth noting that, orthogonal weights by themselves do not guarantee unitary Jacobians. Nonlinear activations still cause the gradients to explode or vanish. Contractive maps such as sigmoid and hyperbolic tangent lead to vanishing gradients. Chen et al. (2018a) offer an initialization scheme taking into account the nonlinearity. However, the theory developed relies heavily on the random matrix assumptions, which only hold at the initialization point of training. Although it has been shown to predict trainability beyond initialization.

**Dynamical systems view of recurrent networks.** Connections between dynamical systems and RNNs have not been well explored. Laurent & von Brecht (2017) study the behavior of dynamical systems induced by recurrent networks, and show that LSTMs and GRUs exhibit chaotic dynamics in the absence of input data. They propose a simplified gated RNN named the chaos free network (CFN), that has non-chaotic dynamics and achieves comparable performance to LSTMs and GRUs on language modeling. Tallec & Ollivier (2018) formulate RNNs as a time-discretized version of ODE and show that time invariance leads to gate-like mechanisms in RNNs. With this formulation,

the authors propose an initialization scheme by setting the initial gate bias according to the range of time dependencies to capture.

**Dynamical systems view of residual networks.** Another line of work that is closely related to ours is the dynamical systems view on residual networks (ResNets) (He et al., 2016). Haber & Ruthotto (2017); Chang et al. (2018a;b); Lu et al. (2018) propose to interpret ResNets as ordinary differential equations (ODEs), under which learning the network parameters is equivalent to solve a parameter estimation problem involving the ODE. Chen et al. (2018b) parameterize the continuous dynamics of hidden units using an ODE specified by a neural network. Stable and reversible architectures are developed (Haber & Ruthotto, 2017; Chang et al., 2018a) from this viewpoint, which form the basis of our proposed recurrent networks.

## 3 ANTISYMMETRICRNNS

### 3.1 ORDINARY DIFFERENTIAL EQUATIONS

We first give a brief overview of the ordinary differential equations (ODEs), a special kind of dynamical systems that involves a single variable, time $t$ in this case. Consider the first-order ODE

$$\boldsymbol{h}'(t) = f(\boldsymbol{h}(t)), \tag{1}$$

for time $t \geq 0$, where $\boldsymbol{h}(t) \in \mathbb{R}^n$ and $f : \mathbb{R}^n \to \mathbb{R}^n$. Together with a given initial condition $\boldsymbol{h}(0)$, the problem of solving for the function $\boldsymbol{h}(t)$ is called the *initial value problem*. For most ODEs, it is impossible to find an analytic solution. Instead, numerical methods relying on discretization are commonly used to approximate the solution. The *forward Euler method* is probably the best known and simplest numerical method for approximation. One way to derive the forward Euler method is to approximate the derivative on the left-hand side of Equation 1 by a finite difference, and evaluate the right-hand side at $\boldsymbol{h}_{t-1}$:

$$\frac{\boldsymbol{h}_t - \boldsymbol{h}_{t-1}}{\epsilon} = f(\boldsymbol{h}_{t-1}). \tag{2}$$

Note that for the approximation to be valid, $\epsilon > 0$ should be small by the definition of the derivative. One can easily prove that the forward Euler method converges linearly w.r.t. $\epsilon$, assuming $f(\boldsymbol{h})$ is Lipschitz continuous on $\boldsymbol{h}$ and that the eigenvalues of the Jacobian of $f$ have negative real parts. Rearranging it, we have the forward Euler method for a given initial value

$$\boldsymbol{h}_t = \boldsymbol{h}_{t-1} + \epsilon f(\boldsymbol{h}_{t-1}), \quad \boldsymbol{h}_0 = \boldsymbol{h}(0). \tag{3}$$

Geometrically, each forward Euler step takes a small step along the tangential direction to the exact trajectory starting at $\boldsymbol{h}_{t-1}$. As a result, $\epsilon$ is usually referred to as the step size.

As an example, consider the ODE

$$\boldsymbol{h}'(t) = \tanh\left(\boldsymbol{W}\boldsymbol{h}(t)\right).$$

The forward Euler method approximates the solution to the ODE iteratively as

$$\boldsymbol{h}_t = \boldsymbol{h}_{t-1} + \epsilon \tanh(\boldsymbol{W}\boldsymbol{h}_{t-1}),$$

which can be regarded as a recurrent network without input data. Here $\boldsymbol{h}_t$ is the hidden state at the $t$-th step, $\boldsymbol{W}$ is a model parameter, and $\epsilon$ is a hyperparameter. This provides a general framework of designing recurrent network architectures by discretizing ODEs. As a result, we can design ODEs that possess desirable properties by exploiting the theoretical successes of dynamical systems, and the resulting recurrent networks will inherit these properties. Stability is one of the important properties to consider, which we will discuss in the next section. It is worth mentioning that the "skip connection" in this architecture resembles the residual RNN (Yue et al., 2018) and the Fourier RNN (Zhang et al., 2018b), which are proposed to mitigate the vanishing and exploding gradient issues.

### 3.2 STABILITY OF ORDINARY DIFFERENTIAL EQUATIONS: ANTISYMMETRICRNNS

In numerical analysis, stability theory addresses the stability of solutions of ODEs under small perturbations of initial conditions. In this section, we are going to establish the connections between

the stability of an ODE and the trainability of the RNNs by discretizing the ODE, and design a new RNN architecture that is stable and capable of capturing long-term dependencies.

An ODE solution is stable if the long-term behavior of the system does not depend significantly on the initial conditions. A formal definition is given as follows.

**Definition 1.** *(Stability) A solution $\boldsymbol{h}(t)$ of the ODE in Equation 1 with initial condition $\boldsymbol{h}(0)$ is stable if for any $\epsilon > 0$, there exists a $\delta > 0$ such that any other solution $\tilde{\boldsymbol{h}}(t)$ of the ODE with initial condition $\tilde{\boldsymbol{h}}(0)$ satisfying $|\boldsymbol{h}(0) - \tilde{\boldsymbol{h}}(0)| \leq \delta$ also satisfies $|\boldsymbol{h}(t) - \tilde{\boldsymbol{h}}(t)| \leq \epsilon$, for all $t \geq 0$.*

In plain language, given a small perturbation of size $\delta$ of the initial state, the effect of the perturbation on the subsequent states is no bigger than $\epsilon$. The eigenvalues of the Jacobian matrix play a central role in stability analysis. Let $\boldsymbol{J}(t) \in \mathbb{R}^{n \times n}$ be the Jacobian matrix of $f$, and $\lambda_i(\cdot)$ denotes the $i$-th eigenvalue.

**Proposition 1.** *The solution of an ODE is stable if*

$$\max_{i=1,2,\ldots,n} Re(\lambda_i(\boldsymbol{J}(t))) \leq 0, \quad \forall t \geq 0, \tag{4}$$

*where $Re(\cdot)$ denotes the real part of a complex number.*

A more precise proposition that involves the kinematic eigenvalues of $\boldsymbol{J}(t)$ is given in Ascher et al. (1994). Stability alone, however, does not suffice to capture long-term dependencies. As argued in Haber & Ruthotto (2017), $Re(\lambda_i(\boldsymbol{J}(t))) \ll 0$ results in a lossy system; the energy or signal in the initial state is dissipated over time. Using such an ODE as the underlying dynamical system of a recurrent network will lead to catastrophic forgetting of the past inputs during the forward propagation. Ideally,

$$Re(\lambda_i(\boldsymbol{J}(t))) \approx 0, \quad \forall i = 1, 2, \ldots, n, \tag{5}$$

a condition we referred to as the *critical criterion*. Under this condition, the system preserves the long-term dependencies of the inputs while being stable.

**Stability and Trainability.** Here we connect the stability of the ODE to the trainability of the RNN produced by discretization. Inherently, the stability analysis studies the sensitivity of a solution, i.e., how much a solution of the ODE would change w.r.t. changes in the initial condition. Differentiating Equation 1 with respect to the initial state $\boldsymbol{h}(0)$ on both sides, we have the following sensitivity analysis (with chain rules):

$$\frac{\mathrm{d}}{\mathrm{d}t}\left(\frac{\partial \boldsymbol{h}(t)}{\partial \boldsymbol{h}(0)}\right) = \boldsymbol{J}(t)\frac{\partial \boldsymbol{h}(t)}{\partial \boldsymbol{h}(0)}. \tag{6}$$

For notational simplicity, let us define $\boldsymbol{A}(t) = \partial \boldsymbol{h}(t)/\partial \boldsymbol{h}(0)$, then we have

$$\frac{\mathrm{d}\boldsymbol{A}(t)}{\mathrm{d}t} = \boldsymbol{J}(t)\boldsymbol{A}(t), \quad \boldsymbol{A}(0) = \boldsymbol{I}. \tag{7}$$

Note that this is a linear ODE with solution $\boldsymbol{A}(t) = e^{\boldsymbol{J} \cdot t} = \boldsymbol{P}e^{\boldsymbol{\Lambda}(\boldsymbol{J})t}\boldsymbol{P}^{-1}$, assuming the Jacobian $\boldsymbol{J}$ does not vary or vary slowly over time (We will later show this is a valid assumption). Here $\boldsymbol{\Lambda}(\boldsymbol{J})$ denotes the eigenvalues of $\boldsymbol{J}$, and the columns of $\boldsymbol{P}$ are the corresponding eigenvectors. See Appendix A for a more detailed derivation. In the language of RNNs, $\boldsymbol{A}(t)$ is the Jacobian of a hidden state $\boldsymbol{h}_t$ with respect to the initial hidden state $\boldsymbol{h}_0$. When the critical criterion is met, i.e., $Re(\boldsymbol{\Lambda}(\boldsymbol{J})) \approx 0$, the magnitude of $\boldsymbol{A}(t)$ is approximately constant in time, thus no exploding or vanishing gradient problems.

With the connection established, we next design ODEs that satisfy the critical criterion. An anti-symmetric matrix is a square matrix whose transpose equals its negative; i.e., a matrix $\boldsymbol{M} \in \mathbb{R}^{n \times n}$ is antisymmetric if $\boldsymbol{M}^T = -\boldsymbol{M}$. An interesting property of an antisymmetric matrix $\boldsymbol{M}$ is that, the eigenvalues of $\boldsymbol{M}$ are all imaginary:

$$Re(\lambda_i(\boldsymbol{M})) = 0, \quad \forall i = 1, 2, \ldots, n,$$

making antisymmetric matrices a suitable building block of a stable recurrent architecture.

Consider the following ODE

$$\boldsymbol{h}'(t) = \tanh\left((\boldsymbol{W}_h - \boldsymbol{W}_h^T)\boldsymbol{h}(t) + \boldsymbol{V}_h\boldsymbol{x}(t) + \boldsymbol{b}_h\right), \tag{8}$$

where $\boldsymbol{h}(t) \in \mathbb{R}^n$, $\boldsymbol{x}(t) \in \mathbb{R}^m$, $\boldsymbol{W}_h \in \mathbb{R}^{n \times n}$, $\boldsymbol{V}_h \in \mathbb{R}^{n \times m}$ and $\boldsymbol{b}_h \in \mathbb{R}^n$. Note that $\boldsymbol{W}_h - \boldsymbol{W}_h^T$ is an antisymmetric matrix. The Jacobian matrix of the right hand side is

$$\boldsymbol{J}(t) = \text{diag} \left[ \tanh' \left( (\boldsymbol{W}_h - \boldsymbol{W}_h^T)\boldsymbol{h}(t) + \boldsymbol{V}_h \boldsymbol{x}(t) + \boldsymbol{b} \right) \right] (\boldsymbol{W}_h - \boldsymbol{W}_h^T), \tag{9}$$

whose eigenvalues are all imaginary, i.e., $Re(\lambda_i(\boldsymbol{J}(t))) = 0, \forall i = 1, 2, \ldots, n$. In other words, it satisfies the critical criterion in Equation 5. See Appendix B for a proof. The entries of the diagonal matrix in Equation 9 are the derivatives of the activation function, which are bounded in $[0, 1]$ for sigmoid and hyperbolic tangent. In other words, the Jacobian matrix $\boldsymbol{J}(t)$ changes smoothly over time. Furthermore, since the input and bias term only affect the bounded diagonal matrix, their effect on the stability of the ODE is insignificant compared with the antisymmetric matrix.

A naive forward Euler discretization of the ODE in Equation 8 leads to the following recurrent network we refer to as the *AntisymmetricRNN*.

$$\boldsymbol{h}_t = \boldsymbol{h}_{t-1} + \epsilon \tanh \left( (\boldsymbol{W}_h - \boldsymbol{W}_h^T)\boldsymbol{h}_{t-1} + \boldsymbol{V}_h \boldsymbol{x}_t + \boldsymbol{b}_h \right), \tag{10}$$

where $\boldsymbol{h}_t \in \mathbb{R}^n$ is the hidden state at time $t$; $\boldsymbol{x}_t \in \mathbb{R}^m$ is the input at time $t$; $\boldsymbol{W}_h \in \mathbb{R}^{n \times n}$, $\boldsymbol{V}_h \in \mathbb{R}^{n \times m}$ and $\boldsymbol{b}_h \in \mathbb{R}^n$ are the parameters of the network; $\epsilon > 0$ is a hyperparameter that represents the step size.

Note that the antisymmetric matrix $\boldsymbol{W}_h - \boldsymbol{W}_h^T$ only has $n(n-1)/2$ degrees of freedom. When implementing the model, $\boldsymbol{W}_h$ can be parameterized as a strictly upper triangular matrix, i.e., an upper triangular matrix of which the diagonal entries are all zero. This makes the proposed model more parameter efficient than an unstructured RNN model of the same size of hidden states.

### 3.3 STABILITY OF THE FORWARD EULER METHOD: DIFFUSION

Given a stable ODE, its forward Euler discretization can still be unstable, as illustrated in Section 4. The stability condition of the forward Euler method has been well studied and summarized in the following proposition.

**Proposition 2.** *(Stability of the forward Euler method) The forward propagation in Equation 10 is stable if*

$$\max_{i=1,2,\ldots,n} |1 + \epsilon \lambda_i(\boldsymbol{J}_t)| \leq 1, \tag{11}$$

*where $|\cdot|$ denote the absolute value or modulus of a complex number and $\boldsymbol{J}_t$ is the Jacobian matrix evaluated at $\boldsymbol{h}_t$.*

See Ascher & Petzold (1998) for a proof. The ODE as defined in Equation 8 is however incompatible with the stability condition of the forward Euler method. Since $\lambda_i(\boldsymbol{J}_t)$, the eigenvalues of the Jacobian matrix, are all imaginary, $|1 + \epsilon \lambda_i(\boldsymbol{J}_t)|$ is always greater than 1, which makes the *AntisymmetricRNN* defined in Equation 10 unstable when solved using forward Euler.

One easy way to fix it is to add *diffusion* to the system by subtracting a small number $\gamma > 0$ from the diagonal elements of the transition matrix. The model thus becomes

$$\boldsymbol{h}_t = \boldsymbol{h}_{t-1} + \epsilon \tanh \left( (\boldsymbol{W}_h - \boldsymbol{W}_h^T - \gamma \boldsymbol{I})\boldsymbol{h}_{t-1} + \boldsymbol{V}_h \boldsymbol{x}_t + \boldsymbol{b}_h \right), \tag{12}$$

where $\boldsymbol{I}$ is the identity matrix of size $n$ and $\gamma > 0$ is a hyperparameter that controls the strength of diffusion. By doing so, the eigenvalues of the Jacobian have slightly negative real parts. This modification improves the stability of the numerical method as demonstrated in Section 4.

### 3.4 GATING MECHANISM

Gating is commonly employed in RNNs. Each gate is often modeled as a single layer network taking the previous hidden state $\boldsymbol{h}_{t-1}$ and data $\boldsymbol{x}_t$ as inputs, followed by a sigmoid activation. As an example, LSTM cells make use of three gates, a forget gate, an input gate, and an output gate. A systematic ablation study suggests that some of the gates are crucial to the performance of LSTM (Jozefowicz et al., 2015).

Gating can be incorporated into AntisymmetricRNN as well. However, it should be done carefully so that the critical condition in Equation 5 still holds. We propose the following modification to

AntisymmetricRNN, which adds an *input gate* $z_t$ to control the flow of information into the hidden states:

$$z_t = \sigma \left( (W_h - W_h^T - \gamma I) h_{t-1} + V_z x_t + b_z \right),$$
$$h_t = h_{t-1} + \epsilon z_t \circ \tanh \left( (W_h - W_h^T - \gamma I) h_{t-1} + V_h x_t + b_h \right), \quad (13)$$

where $\sigma$ denotes the sigmoid function and $\circ$ denotes the Hadamard product.

The effect of $z_t$ resembles the input gate in LSTM and the update gate in GRU. By sharing the antisymmetric weight matrix, the number of model parameters only increases slightly, instead of being doubled. More importantly, the Jacobian matrix of this gated model has a similar form as that in Equation 9, that is, a diagonal matrix multiplied by an antisymmetric matrix (ignoring diffusion). As a result, the real parts of the eigenvalues of the Jacobian matrix are still close to zero, and the critical criterion remains satisfied.

## 4 SIMULATION

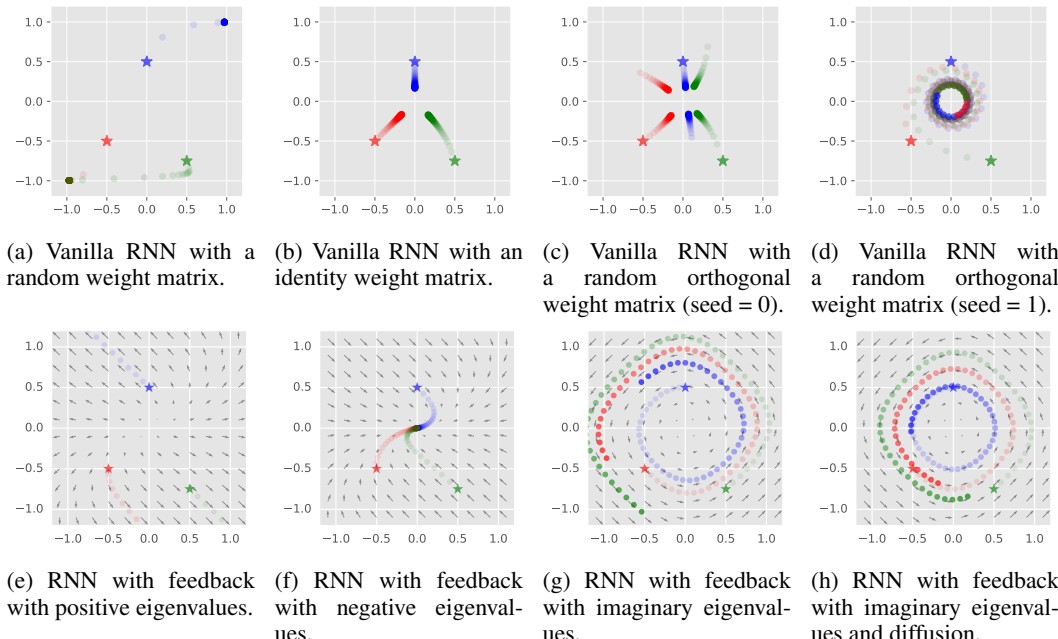

(a) Vanilla RNN with a random weight matrix.

(b) Vanilla RNN with an identity weight matrix.

(c) Vanilla RNN with a random orthogonal weight matrix (seed = 0).

(d) Vanilla RNN with a random orthogonal weight matrix (seed = 1).

(e) RNN with feedback with positive eigenvalues.

(f) RNN with feedback with negative eigenvalues.

(g) RNN with feedback with imaginary eigenvalues.

(h) RNN with feedback with imaginary eigenvalues and diffusion.

Figure 1: Visualization of the dynamics of RNNs and RNNs with feedback using different weight matrices.

Adopting the visualization technique used by Laurent & von Brecht (2017) and Haber & Ruthotto (2017), we study the behavior of two-dimensional vanilla RNNs (left) and RNNs with feedback (right) in the absence of input data and bias:

$$\text{vanilla: } h_t = \tanh(W h_{t-1}), \qquad \text{feedback: } h_t = h_{t-1} + \epsilon \tanh(W h_{t-1}).$$

Here $h_t \in \mathbb{R}^2$ and $1 \le t \le T$. We arbitrarily choose three initial states: $(0, 0.5), (-0.5, -0.5)$ and $(0.5, -0.75)$, and apply the corresponding RNN recurrence. Figure 1 plots the progression of the states $h_t$ of vanilla RNNs (first row) and RNNs with feedback (second row) parameterized by different transition matrices $W$. In each figure, more translucent points represent earlier time steps. The number of total time steps $T = 50$.

Figure 1(a) corresponds to a random weight matrix, where the entries are independent and standard Gaussian. The three initial points (shown in stars) converge to two fixed points on the boundary. Since the weight matrix is unstructured, the behavior is unpredictable as expected. More examples with different weight matrices are shown in Appendix D. Figure 1(b) shows the behavior of the

identity weight matrix. Since $\tanh(\cdot)$ is a contractive mapping, the origin is the unique fixed point. Figure 1(c) and (d) correspond to orthogonal weight matrices that represent reflection and rotation transforms respectively. A two-dimensional orthogonal matrix is either a reflection or a rotation transform; the determinant of the matrix is 1 or $-1$ respectively. Even though the weight matrix is orthogonal, the states all converge to the origin because of the derivative of the activation function $0 \le \tanh'(\boldsymbol{W}\boldsymbol{h}_{t-1}) \le 1$.

In the case of RNNs with feedback, the trajectory of the hidden states is predictable based on the eigenvalues of the weight matrix $\boldsymbol{W}$. We consider the following four weight matrices that correspond to Figure 1(e)-(f) respectively:

$$\boldsymbol{W}_+ = \begin{pmatrix} 2 & -2 \\ 0 & 2 \end{pmatrix}, \boldsymbol{W}_- = \begin{pmatrix} -2 & 2 \\ 0 & -2 \end{pmatrix}, \boldsymbol{W}_0 = \begin{pmatrix} 0 & -2 \\ 2 & 0 \end{pmatrix}, \boldsymbol{W}_{\mathrm{diff}} = \begin{pmatrix} -0.15 & -2 \\ 2 & -0.15 \end{pmatrix}.$$

We also overlay the vector field that represents the underlying ODE. The step size is set to $\epsilon = 0.1$.

For Figure 1(e) and (f), the eigenvalues are $\lambda_1(\boldsymbol{W}_+) = \lambda_2(\boldsymbol{W}_+) = 2$ and $\lambda_1(\boldsymbol{W}_-) = \lambda_2(\boldsymbol{W}_-) = -2$. As a result, the hidden states are moving away from the origin and towards the origin respectively. Figure 1(g) corresponds to the antisymmetric weight matrix $\boldsymbol{W}_0$, whose eigenvalues are purely imaginary: $\lambda_1(\boldsymbol{W}_0) = 2i$, $\lambda_2(\boldsymbol{W}_0) = -2i$. In this case, the vector field is circular; a state moves around the origin without exponentially increasing or decreasing its norm. However, on closer inspection, the trajectories are actually outward spirals. This is because, at each time step, the state moves along the tangential direction by a small step, which increases the distance from the origin and leads to numerical instability. It is the behavior characterized by Proposition 2. This issue can be mitigated by subtracting a small diffusion term $\gamma$ from the diagonal elements of the weight matrix. We choose $\gamma = 0.15$ and the weight matrix $\boldsymbol{W}_{\mathrm{diff}}$ has eigenvalues of $\lambda_1(\boldsymbol{W}_{\mathrm{diff}}) = -0.15 + 2i$, $\lambda_2(\boldsymbol{W}_{\mathrm{diff}}) = -0.15 - 2i$. Figure 1(h) shows the effect of the diffusion terms. The vector field is slightly tilting toward the origin and the trajectory maintains a constant distance from the origin.

These simulations show that the hidden states of an AntisymmetricRNN (Figure 1(g) and (h)) have predictable dynamics. It achieves the desirable behavior, without the complication of maintaining an orthogonal or unitary matrix as in Figure 1(d), which still suffers from vanishing gradients due to the contraction of the activation function.

## 5 EXPERIMENTS

The performance of the proposed antisymmetric networks is evaluated on four image classification tasks with long-range dependencies. The classification is done by feeding pixels of the images as a sequence to RNNs and sending the last hidden state $\boldsymbol{h}_T$ of the RNNs into a fully-connected layer and a softmax function. We use the cross-entropy loss and SGD with momentum and Adagrad (Duchi et al., 2011) as optimizers. More experimental details can be found in Appendix C. In this section, *AntisymmetricRNN* denotes the model with diffusion in Equation 12, and *AntisymmetricRNN w/ gating* represents the model in Equation 13.

### 5.1 PIXEL-BY-PIXEL MNIST

In the first task, we learn to classify the MNIST digits by pixels (LeCun et al., 1998). This task was proposed by Le et al. (2015) and used as a benchmark for learning long term dependencies. MNIST images are grayscale with $28 \times 28$ pixels. The 784 pixels are presented sequentially to the recurrent net, one pixel at a time in scanline order (starting at the top left corner of the image and ending at the bottom right corner). In other words, the input dimension $m = 1$ and number of time steps $T = 784$. The *pixel-by-pixel MNIST* task is to predict the digit of the MNIST image after seeing all 784 pixels. As a result, the network has to be able to learn the long-range dependencies in order to correctly classify the digit. To make the task even harder, the MNIST pixels are shuffled using a fixed random permutation. It creates non-local long-range dependencies among pixels in an image. This task is referred to as the *permuted pixel-by-pixel MNIST*.

---

[1]Cooijmans et al. (2017) also report the performance of the LSTM as a baseline: 98.9% on MNIST and 90.2% on pMNIST. We decide to use the LSTM baseline reported by Arjovsky et al. (2016) because it has a higher accuracy on the more challenging pMNIST task than that in Cooijmans et al. (2017) (92.6% vs 90.2%).

| method | MNIST | pMNIST | # units | # params |
|---|---|---|---|---|
| LSTM (Arjovsky et al., 2016)[1] | 97.3% | 92.6% | 128 | 68k |
| FC uRNN (Wisdom et al., 2016) | 92.8% | 92.1% | 116 | 16k |
| FC uRNN (Wisdom et al., 2016) | 96.9% | 94.1% | 512 | 270k |
| Soft orthogonal (Vorontsov et al., 2017) | 94.1% | 91.4% | 128 | 18k |
| KRU (Jose et al., 2017) | 96.4% | 94.5% | 512 | 11k |
| **AntisymmetricRNN** | 98.0% | **95.8%** | 128 | 10k |
| **AntisymmetricRNN w/ gating** | **98.8%** | 93.1% | 128 | 10k |

Table 1: Evaluation accuracy on pixel-by-pixel MNIST and permuted MNIST.

Table 1 summarizes the performance of our methods and the existing methods. On both tasks, the proposed AntisymmetricRNNs outperform the regular LSTM model using only $1/7$ of parameters. Imposing orthogonal weights (Arjovsky et al., 2016) produces worse results, which corroborates with the existing study showing that such constraints restrict the capacity of the learned model. Softening the orthogonal weights constraints (Wisdom et al., 2016; Vorontsov et al., 2017; Jose et al., 2017) leads to slightly improved performance. AntisymmetricRNNs outperform these methods by a large margin, without the computational overhead to enforce orthogonality.

## 5.2 PIXEL-BY-PIXEL CIFAR-10

To test our methods on a larger dataset, we conduct experiments on *pixel-by-pixel CIFAR-10*. The CIFAR-10 dataset contains $32 \times 32$ colour images in 10 classes (Krizhevsky & Hinton, 2009). Similar to pixel-by-pixel MNIST, we feed the three channels of a pixel into the model at each time step. The input dimension $m = 3$ and number of time steps $T = 1024$.

The results are shown in Table 2. The AntisymmetricRNN performance is on par with the LSTM, and AntisymmetricRNN with gating is slightly better than LSTM, both using only about half of the parameters of the LSTM model. Further investigation shows that the task is mostly dominated by short term dependencies. LSTM can achieve about 48.3% classification accuracy by only seeing the last 8 rows of CIFAR-10 images [2].

| method | pixel-by-pixel | noise padded | # units | # params |
|---|---|---|---|---|
| LSTM | 59.7% | 11.6% | 128 | 69k |
| Ablation model | 54.6% | 46.2% | 196 | 42k |
| **AntisymmetricRNN** | 58.7% | 48.3% | 256 | 36k |
| **AntisymmetricRNN w/ gating** | **62.2%** | **54.7%** | 256 | 37k |

Table 2: Evaluation accuracy on pixel-by-pixel CIFAR-10 and noise padded CIFAR-10.

## 5.3 NOISE PADDED CIFAR-10

To introduce more long-range dependencies to the pixel-by-pixel CIFAR-10 task, we define a more challenging task call the *noise padded CIFAR-10*, inspired by the noise padded experiments in Chen et al. (2018a). Instead of feeding in one pixel at one time, we input each row of a CIFAR-10 image at every time step. After the first 32 time steps, we input independent standard Gaussian noise for the remaining time steps. Since a CIFAR-10 image is of size 32 with three RGB channels, the input dimension is $m = 96$. The total number of time steps is set to $T = 1000$. In other words, only the first 32 time steps of input contain salient information, all remaining 968 time steps are merely random noise. For a model to correctly classify an input image, it has to remember the information from a long time ago. This task is conceptually more difficult than the pixel-by-pixel CIFAR-10, although the total amount of signal in the input sequence is the same. The results are shown in Table 2. LSTM fails to train at all on this task while our proposed methods perform reasonably well with fewer parameters.

---

[2]Last one row: 33.6%, last two rows: 35.6%, last four rows: 39.6%, last eight rows: 48.3%.

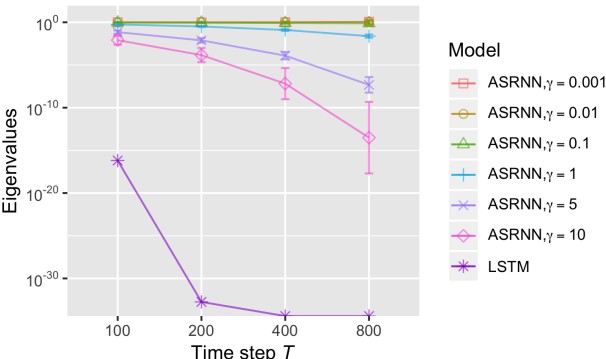

Figure 2: Mean and standard deviation of eigenvalues of the end-to-end Jacobian matrix in Antisymmetric RNNs with different diffusion constants and LSTMs, trained on the noise padded CIFAR-10.

To verify that AntisymmetricRNNs indeed mitigate the exploding/vanishing gradient issues, we conduct an additional set of experiments varying the length of noise padding so that the total time steps $T \in \{100, 200, 400, 800\}$. LSTMs and AntisymmetricRNNs with different diffusion constants $\gamma \in \{0.001, 0.01, 0.1, 1, 5, 10\}$ are trained on these tasks. Figure 2 visualizes the mean and standard deviation of the eigenvalues of the end-to-end Jacobian matrices for these networks. Unitary eigenvalues, i.e., mean close to 1 and standard deviation close to 0, indicate non-exploding and non-vanishing gradients. As shown in the figure, the eigenvalues for LSTMs quickly approaches zero as time steps increase, indicating vanishing gradients during back-propagation. This explains why LSTMs fail to train at all on this task. AntisymmetricRNNs with a broad range of diffusion constants $\gamma$, on the other hand, have eigenvalues centered around 1. It is worth noting though as the diffusion constant increases to large values, AntisymmetricRNNs run into vanishing gradients as well. The diffusion constant $\gamma$ plays an important role in striking a balance between the stability of discretization and capturing long-term dependencies.

### 5.4 ABLATION STUDY

For the CIFAR-10 experiments, we have also conducted an ablation study to further demonstrate the effect of antisymmetric weight matrix. The *ablation model* in Table 2 refers to the model that replaces the antisymmetric weight matrices in Equation 13 with unstructured weight matrices. As shown in Table 2, without the antisymmetric parametrization, the performance on both pixel-by-pixel and noise padded CIFAR-10 is worse.

It is worth mentioning that the antisymmetric formulation is a sufficient condition of stability, not necessary. There are possibly other conditions that lead to stability as well as suggested in Chen et al. (2018a), which could explain the modest degradation in the ablation results.

### 6 CONCLUSION

In this paper, we present a new perspective on the trainability of RNNs from the dynamical system viewpoint. We draw connections between RNNs and the ordinary differential equation theory and design new recurrent architectures by discretizing ODEs. This new view opens up possibilities to exploit the computational and theoretical success from dynamical systems to understand and improve the trainability of RNNs. We also propose the AntisymmetricRNN, which is a discretization of ODEs that satisfy the critical criterion. Besides its appealing theoretical properties, our models have demonstrated competitive performance over strong recurrent baselines on a comprehensive set of benchmark tasks.

By establishing a link between recurrent networks and ordinary differential equations, we anticipate that this work will inspire future research in both communities. An important item of future work is to investigate other stable ODEs and numerical methods that lead to novel and well-conditioned recurrent architectures.

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

## A    AN OVERVIEW OF STABILITY THEORY

In this section, we provide a brief overview of the stability theory by examples. Most of the materials are adapted from Ascher & Petzold (1998).

Consider the simple scalar ODE, often referred to as the *test equation*: $y'(t) = \lambda y(t)$, where $\lambda \in \mathbb{C}$ is a constant. We allow $\lambda$ to be complex because it later represents an eigenvalue of a system's matrix. The solution to this initial value problem is $y(t) = e^{\lambda t} y(0)$. If $y(t)$ and $\tilde{y}(t)$ are two solutions of the test equation, then their difference at any time $t$ is

$$|y(t) - \tilde{y}(t)| = |e^{\lambda t}(y(0) - \tilde{y}(0))| = e^{Re(\lambda)t}|y(0) - \tilde{y}(0)|. \tag{14}$$

If $Re(\lambda) > 0$, then a small perturbation of the initial states would cause an exponentially exploding difference. If $Re(\lambda) < 0$, the system is stable, but the perturbation decays exponentially. The perturbation is preserved in the system only if $Re(\lambda) = 0$.

We now consider the extension of the test equation to a matrix ODE $\boldsymbol{y}'(t) = \boldsymbol{A}\boldsymbol{y}(t)$. The solution is $\boldsymbol{y}(t) = e^{\boldsymbol{A}t}\boldsymbol{y}(0)$. To simplify the analysis, we assume $\boldsymbol{A}$ is diagonalizable, i.e., $\boldsymbol{P}^{-1}\boldsymbol{A}\boldsymbol{P} = \boldsymbol{\Lambda}$, where $\boldsymbol{\Lambda}$ is a diagonal matrix of the eigenvalues of $\boldsymbol{A}$ and the columns of $\boldsymbol{P}$ are the corresponding eigenvectors. If we define $\boldsymbol{w}(t) = \boldsymbol{P}^{-1}\boldsymbol{y}(t)$, then $\boldsymbol{w}'(t) = \boldsymbol{\Lambda}\boldsymbol{w}(t)$. The system for $\boldsymbol{w}(t)$ is decoupled: for each component $w_i(t)$ of $\boldsymbol{w}(t)$, we have a test equation $w_i'(t) = \lambda_i w_i(t)$. Therefore, the stability for $\boldsymbol{w}(t)$, hence also for $\boldsymbol{y}(t)$, is determined by the eigenvalues $\lambda_i$.

## B    PROOF OF A PROPOSITION

In this section, we provide a proof of a proposition which implies that the AntisymmetricRNN and AntisymmetricRNN with gating satisfy the critical criterion, i.e., the eigenvalues of the Jacobian matrix are imaginary. The proof is adapted from Chang et al. (2018a).

**Proposition 3.** *If $\boldsymbol{W} \in \mathbb{R}^{n \times n}$ is an antisymmetric matrix and $\boldsymbol{D} \in \mathbb{R}^{n \times n}$ is an invertible diagonal matrix, then the eigenvalues of $\boldsymbol{DW}$ are imaginary.*

$$Re(\lambda_i(\boldsymbol{DW})) = 0, \quad \forall i = 1, 2, \ldots, n.$$

*Proof.* Let $\lambda$ and $\boldsymbol{v}$ be a pair of eigenvalue and eigenvector of $\boldsymbol{DW}$, then

$$\boldsymbol{DW}\boldsymbol{v} = \lambda\boldsymbol{v},$$
$$\boldsymbol{W}\boldsymbol{v} = \lambda\boldsymbol{D}^{-1}\boldsymbol{v},$$
$$\boldsymbol{v}^*\boldsymbol{W}\boldsymbol{v} = \lambda(\boldsymbol{v}^*\boldsymbol{D}^{-1}\boldsymbol{v}),$$

On one hand, $\boldsymbol{v}^*\boldsymbol{D}^{-1}\boldsymbol{v}$ is real. On the other hand,

$$(\boldsymbol{v}^*\boldsymbol{W}\boldsymbol{v})^* = \boldsymbol{v}^*\boldsymbol{W}^*\boldsymbol{v} = -\boldsymbol{v}^*\boldsymbol{W}\boldsymbol{v},$$

where $*$ represents conjugate transpose. It implies that $\boldsymbol{v}^*\boldsymbol{W}\boldsymbol{v}$ is imaginary. Therefore, $\lambda$ has to be imaginary. As a result, all eigenvalues of $\boldsymbol{D}\boldsymbol{W}$ are imaginary. □

## C    EXPERIMENTAL DETAILS

Let $m$ be the input dimension and $n$ be the number of hidden units. The input to hidden matrices are initialized to $\mathcal{N}(0, 1/m)$. The hidden to hidden matrices are initialized to $\mathcal{N}(0, \sigma_w^2/n)$, where $\sigma_w$ is chosen from $\sigma_w \in \{0, 1, 2, 4, 8, 16\}$. The bias terms are initialized to zero, except the forget gate bias of LSTM is initialized to 1, as suggested by Jozefowicz et al. (2015). For AntisymmetricRNNs, the step size $\epsilon \in \{0.01, 0.1, 1\}$ and diffusion $\gamma \in \{0.001, 0.01, 0.1, 1.0\}$. We use SGD with momentum and Adagrad (Duchi et al., 2011) as optimizers, with batch size of 128 and learning rate chosen from $\{0.1, 0.2, 0.3, 0.4, 0.5, 0.75, 1\}$. On MNIST and pixel-by-pixel CIFAR-10, all the models are trained for 50,000 iterations. On noise padded CIFAR-10, models are trained for 10,000 iterations. We use the standard train/test split of MNIST and CIFAR-10. The performance measure is the classification accuracy evaluated on the test set.

## D    ADDITIONAL VISUALIZATIONS

In this section, we present additional visualizations that are related to the simulation study in Section 4. Figure 3 and 4 show the dynamics of vanilla RNNs and AntisymmetricRNNs with standard Gaussian random weights using different seeds. These visualizations further illustrate the random behavior of a vanilla RNN and the predictable dynamics of an AntisymmetricRNN.

Figure 5 shows the dynamics of AntisymmetricRNNs with independent standard Gaussian input. This shows that the dynamics become noisier compared to Figure 1, but the trend remains the same.

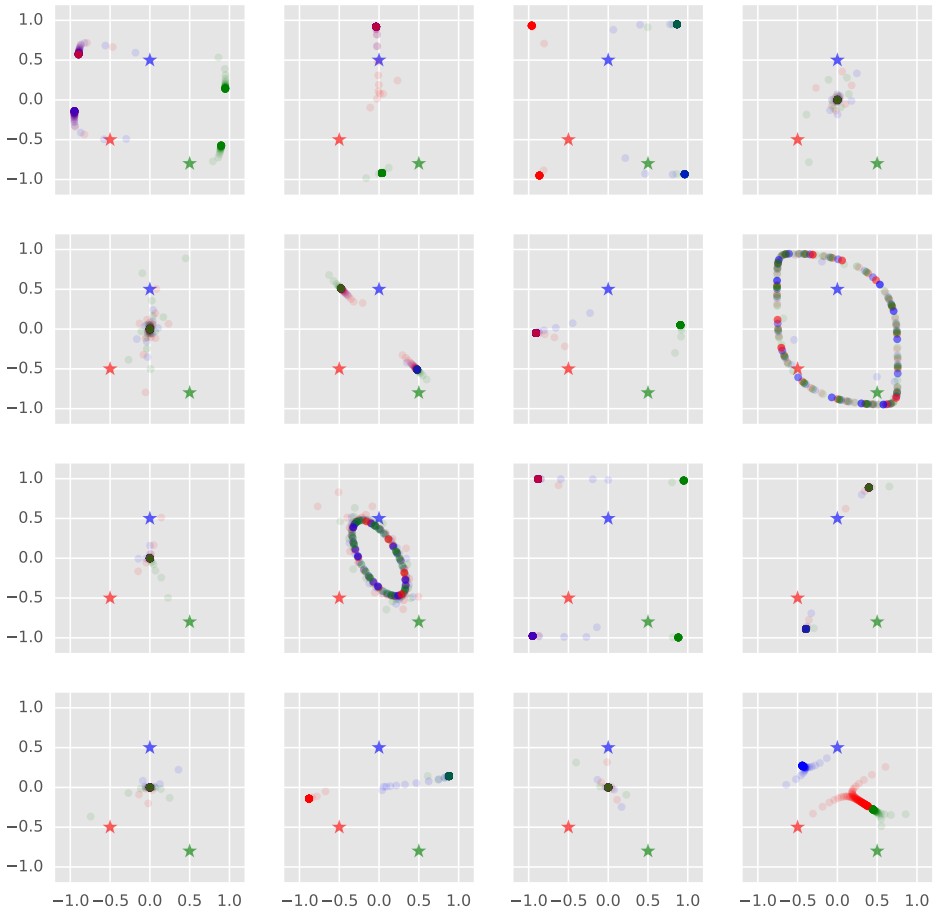

Figure 3: Visualization of the dynamics of vanilla RNNs with standard Gaussian random weights, using seeds from 1 to 16.

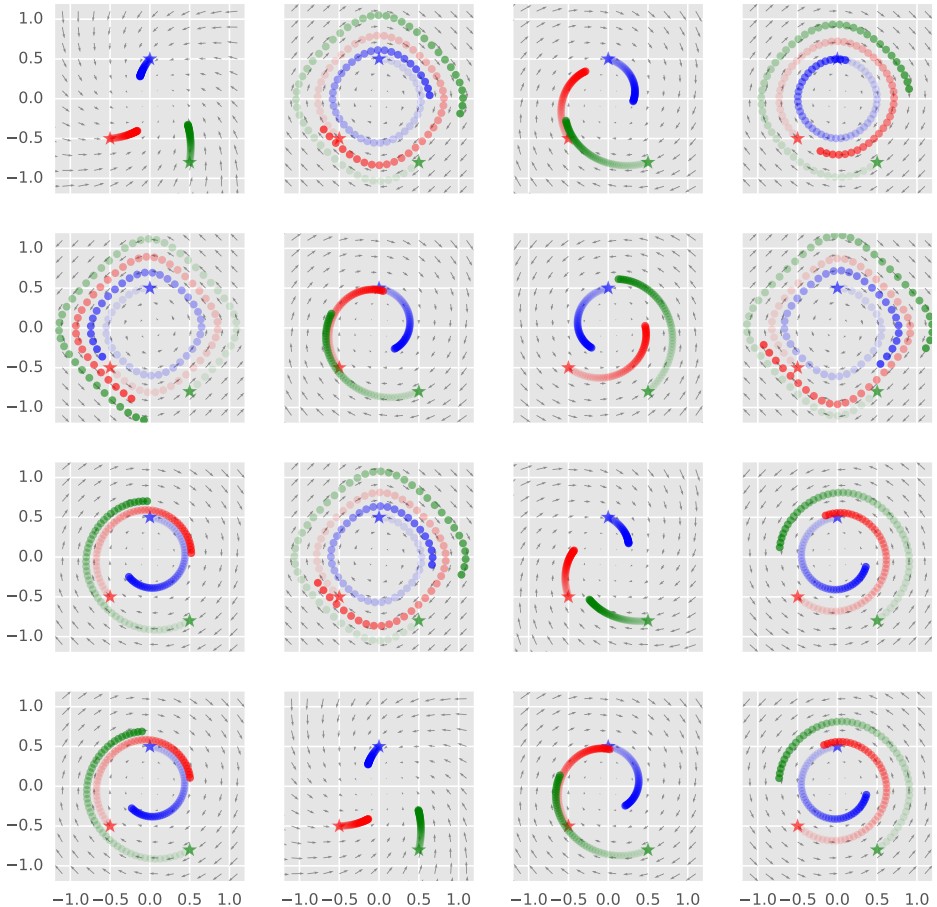

Figure 4: Visualization of the dynamics of RNN with feedback with standard Gaussian random weights, using seeds from 1 to 16, diffusion strength $\gamma = 0.1$.

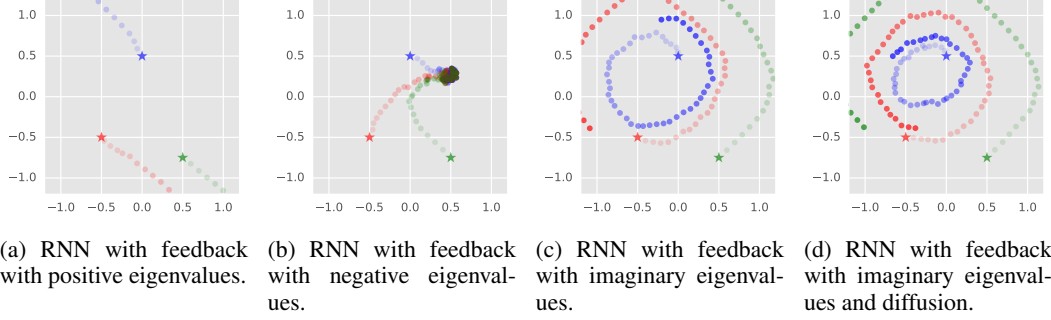

(a) RNN with feedback with positive eigenvalues.

(b) RNN with feedback with negative eigenvalues.

(c) RNN with feedback with imaginary eigenvalues.

(d) RNN with feedback with imaginary eigenvalues and diffusion.

Figure 5: Visualization of the dynamics of RNN with feedback with independent standard Gaussian input.

