# OpenReview forum: "AntisymmetricRNN: A Dynamical System View on Recurrent Neural Networks"
_ICLR.cc/2019/Conference_

### Official Review · AnonReviewer3 · 2018-10-31
**capacity of long-term storage?**

**Rating:** 6
**Confidence:** 5

**Review:**

This is an interesting paper which proposes a novel angle on the problem of learning long-term dependencies in recurrent nets. The authors argue that most of the action should be in the imaginary part of the eigenvalues of the Jacobian J=F' of the new_state = old_state + epsilon F(old_state, input) incremental type of recurrence, while the real part should be slightly negative. If they were 0 the discrete time updates would still not be stable, so slightly negative (which leads to exponential loss of information) leads to stability while making it possible for the information decay to be pretty slow. They also propose a gated variant which sometimes works better.

This is similar to earlier work based on orthogonal or unitary Jacobians of new_state = H(old_state,input) updates, since the Jacobian of H(old_state,input) = old_state + epsilon F( old_state,input) is I + epsilon F'. In this light, it is not clear why the proposed architecture would be better than the partially orthogonal / unitary variants previously proposed. My general concern with this this type of architecture is that they can store information in 'cycles' (like in fig 1g, 1h) but this is a pretty strong constraint. For example, in the experiments, the authors did not apparently vary the length of the sequences (which would break the trick of using periodic attractors to store information). In practical applications this is very important. Also, all of the experiments are with classification tasks with few categories (10), i.e., requiring only storing 4 bits of information. Memorization tasks requiring to store many more bits, and with randomly varying sequence lengths, would better test the abilities of the proposed architecture.

---

> ### Author Response · Authors · 2018-11-15
> **Response to AnonReviewer3**
>
> Thank you for your comments and feedback.
>
> “Connection to prior work on orthogonal/unitary weights”? Thanks to the reviewer for bringing up another angle to connect this work to the prior work on orthogonal/unitary weights. While prior work reaches unitary Jacobian by constraining the weight matrices to be orthogonal/unitary with linear activation (the condition breaks if nonlinear activation is used), unitary Jacobian is reached in AntisymmetricRNN with the residual connection and constraining f’ to have imaginary eigenvalues. Unitary/orthogonal matrices have eigenvalues that lie on the unit circle. Antisymmetric matrices have eigenvalues of the form i\lambda where \lambda is arbitrary. This implies that the dimension of the possible transformation is much larger (the whole imaginary axis). Therefore, antisymmetric networks are more expressive than unitary ones. There are three advantages of our approach: 1) our condition can be easily achieved with the antisymmetric weight parameterization, with no computational overhead; 2) our condition takes nonlinear activations into consideration; 3) we empirically demonstrate that our formulation is more expressive than constraining the weight matrix to be orthogonal/unitary, as shown in Table 1. Moreover, we expect the connections between RNNs and the ODE theory to serve as a framework to inspire new RNN architectures in the future.
>
> “store information in 'cycles'”. The behavior of the network in phase space is not repetitive. Similar manifolds are obtained when one looks at Lorenz systems for example, which is a simplification of the weather system. The phase diagrams suggest that the network never blows or decays but it is important to note that it does not repeat itself and samples different points in space.
>
> We thank the reviewer for suggesting the tasks with more categories and varying sequence lengths. It is definitely worth studying the performance of AntisymmetricRNN on tasks such as copy and addition in future work.

---

### Official Review · AnonReviewer1 · 2018-11-02
**Good paper with original work, experiments could be improved**

**Rating:** 7
**Confidence:** 5

**Review:**

In this paper, the authors provide a new approach to analyze the behavior of
RNNs by relating the RNNs with ODE numerical schemes. They provide analysis on
the stability of forward Euler scheme and proposed an RNN architecture called
AntisymmetricRNN to solve the gradient exploding/vanishing problem.

The paper is well presented although more recent works in this direction
should be cited and discussed. Also, some important issues are omitted and not
explained.
For example, the analysis begins with "RNNs with feedback" rather than vanilla
RNN, since vanilla RNN does not have the residual structure as eq(3). The
authors should note that clearly in the paper.

Although there are previous works relating ResNets with ODEs, such as [1],
this paper is original as it is the first work that relates the stability of
ODE numerical scheme with the gradient vanishing/exploding issues in RNNs.

In general, this paper provides a novel approach to analyze the gradient
vanishing/exploding issue in RNNs and provides applicable solutions, thus I
recommend to accept it.


Detailed comments:

The gradient exploding/vanishing issue has been extensively studied these
years and more recent results should be discussed in related works.
Author mentioned that existing methods "come with significant computational
overhead and reportedly hinder representation power of these models". However
this is not true for [2] which achieves full expressive power with
no-overhead.
It is true that "orthogonal weight matrices alone does not prevent exploding
and vanishing gradients", thus there are architectural approaches that can
bound the gradient norm by constants [3].

The authors argued that the critical criterion is important in preserving the
gradient norm. However, later on added a diffusion term to maintain the
stability of forward Euler method. Thus the gradient will vanish
exponentially w.r.t. time step t as: (1-\gamma)^t. Could the authors provide
more detailed analysis on this issue?

Since eq(3) cannot be regarded as vanilla RNN, it would be better begin the
analysis with advanced RNN architectures that fit in this form, such as
Residual RNN, Statistical Recurrent Units and Fourier Recurrent Units.

Why sharing the weight matrix of gated units and recurrent units? Is there any
other reason to do this other than reducing the number of parameters?

More experiment should be conducted on real applications of RNN, such as
language model or machine translation.


[1] Yiping Lu, Aoxiao Zhong, Quanzheng Li, and Bin Dong. Beyond finite layer
neural networks: Bridging deep architectures and numerical differential
equations. In ICML, pp. 3276–3285, 2018.

[2] Zhang, Jiong, Qi Lei, and Inderjit S. Dhillon. "Stabilizing Gradients for
Deep Neural Networks via Efficient SVD Parameterization." In ICML, pp.
5806-5814, 2018.

[3] Zhang, Jiong, Yibo Lin, Zhao Song, and Inderjit S. Dhillon. "Learning Long
Term Dependencies via Fourier Recurrent Units." In ICML, pp. 5815-5823, 2018.

---

> ### Author Response · Authors · 2018-11-15
> **Response to AnonReviewer1**
>
> Thank you for the detailed comments and for pointing us to the latest work on the Spectral RNN and Fourier Recurrent Units. We have updated the paper accordingly.
>
> “diffusion breaks the critical criterion”? We want to emphasize that the critical criterion describes a condition of stability of the underlying ODE w.r.t. initial values, while the diffusion term is necessary to stabilize the forward Euler discretization of the ODE. The AntisymmetricRNN does run into issues of vanishing gradient when the diffusion factor is set to large values, with order (1-c\epsilon\gamma)^t. Here \epsilon is the step size of Euler discretization, \gamma is the diffusion factor and c captures the derivatives of the hidden activation. Due to the small step size and the bounded derivatives, we find AntisymmetricRNN can tolerate a broad range of diffusion factors, as shown in the new Figure 2 added on the eigenvalues.
>
> “begin the analysis with advanced RNN architectures that fit in this form”. We have added discussion of more advanced recurrent architectures that fit in the “residual connection” form.
>
> “Why sharing the weight matrix of gated units and recurrent units”? The weight matrix is shared between the gated units and recurrent units to satisfy the critical criterion. When the weight matrix is shared, the Jacobian matrix has the form of (D_1 + D_2) M, where D_1 and D_2 are diagonal matrices and M is an antisymmetric matrix. On the other hand, if the gated units and recurrent units use different weights, then the Jacobian matrix has the form of D_1 M_1 + D_2 M_2. Even if both M_1 and M_2 are antisymmetric, the eigenvalues of the Jacobian matrix can have real parts, thus breaking the criticality.
>
> “Conduct experiment on language models and machine translation”? We conducted experiments on the pixel-by-pixel image tasks as the benchmark datasets for studying long-range dependence to demonstrate the effectiveness of the proposed method. We would like to study the performance of AntisymmetricRNN on language models and machine translation in future work.

---

### Official Review · AnonReviewer2 · 2018-11-02
**A novel RNN architecture**

**Rating:** 7
**Confidence:** 5

**Review:**

This paper introduces antisymmetric RNN, a novel RNNs architecture that is motivated through ordinary differential equation (ODE) framework. Authors consider a first order ODE and the RNN that results from the discretization of this ODE. They show how the stability criteria with respect to perturbation of  the initial state results in an ODE  in a trainability criteria for the corresponding  RNN. This criteria ensures that there are  no exploding/vanishing gradients. Authors then propose a specific parametrization, relying on antisymmetric matrix to ensure that the stability/trainability criteria is respected. They also propose a gated-variant of their architecture.  Authors evaluate their proposal on pixel-by-pixel MNIST and CIFAR10 where they show they can outperforms an LSTM.

The paper is well-written and pleasant to read. However, while the authors argue that their architecture allows to mitigate vanishing/exploding gradient, there is no empirically verification of this claim. In particular, it would be nice to visualize how the gradient norm changes as the gradient is  backpropagated in time, compare the gradient flows of Antisymmetric RNN with a LSTM or report the top eigenvalue of the jacobian for the different models.

In addition,  the analysis for the antisymmetric RNN assumes no input is given to the model. It is not clear to me how having an input at each timestep affects those results?

A few more specific questions/remarks:
-	Experimentally, authors find that the gated antisymmetric RNN sometime outperforms its non-gated counterpart. However, one motivation for the gate mechanism is to better control the gradients flow. It is unclear to me what is the motivation of using gate for the antisymmetric RNN ?
-	as the proposed RNN relies on a antisymmetric matrix to represent the hidden-to-hidden transition matrix, which has less degree of liberty, can we expect the antisymmetric RNN to have same expressivity as a standard RNN. In particular, how easily can an antisymmetric RNN forgets information ?
-	On the pixel-by-pixel MNIST, authors report the Arjosky results for the LSTM baseline.
Note that some papers reported better performance for the LSTM baseline such as Recurrent Batch Norm (Cooijman et al., 2016) .

Antisymmetric RNN appears to be well-motived architecture and seems to outperforms previous RNN variants that also aims at solving exploding/vanishing gradient problem. Overall I lean toward acceptance, although I do think that adding an experiment explicitly showing that the gradient does not explode/vanish would strengthen the paper.


* Revision

Thanks for your response,  the paper new  version address my main concerns, I appreciate the new experiment looking at the eigenvalues of the  end-to-end Jacobian which clearly shows the advantage of the AntisymmetricRNN.

---

> ### Author Response · Authors · 2018-11-15
> **Response to AnonReviewer2**
>
> Thank you for your constructive feedback. The paper is updated with the suggested changes. In particular, a new Figure 2 visualizing the eigenvalues of the end-to-end Jacobian is included.
>
> “empirical verification of mitigation of vanishing/exploding gradient”? We included a new Figure 2 on the mean and standard deviation of the eigenvalues of the end-to-end Jacobian matrices for LSTMs and AntisymmetricRNNs with different diffusion constants. They are computed on the networks trained for the padded CIFAR10 dataset with time steps T in {100, 200, 400, 800}. As a quick summary, the eigenvalues for LSTMs quickly approaches zero as time steps increase, indicating vanishing gradients as they back-propagate in time. This explains why LSTMs fail to train at all on this task. AntisymmetricRNNs with a broad range of diffusion, on the other hand, have eigenvalues centered around 1. It is worth noting though as the diffusion constant increases to large values, AntisymmetricRNNs run into vanishing gradients as well. The diffusion constant plays an important role in striking a balance between the stability of discretization and non-vanishing gradients.
>
> “how do inputs affects the analysis”? Our analysis in Section 3 on the stability of an ODE is valid with inputs. In Equation 9 where we calculate the Jacobian matrix, the inputs only affect the diagonal matrix. As long as the diagonal matrix is bounded, which is true for derivatives of most activation functions, the Jacobian matrix still satisfies the critical criterion with inputs. Figure 5 in Appendix D shows the simulation with independent standard Gaussian input. Although the dynamics become slightly noisier comparing with those in Figure 1, the trend remains the same.
>
> “the motivation of using gate for the antisymmetric RNN”? We see AntisymmetricRNN and AntisymmetricRNN w/ gating as discretizations of two different ODEs under the same theoretical framework. Gating provides a mechanism for the underlying ODE to have more degrees of freedom and to capture more complex dynamics. Experimental results show that AntisymmetricRNN performs better on pMNIST while AntisymmetricRNN w/ gating works well on the other tasks.
>
> “expressivity of AntisymmetricRNN”? Structural constraint on the weight matrix could limit the expressivity of AntisymmetricRNN. However, we do not observe performance degradation in our empirical studies. We hypothesize it is due to over-parametrization in these networks. An AntisymmetricRNN can outperform other RNN models with fewer model parameters.
>
> “how easily can an antisymmetric RNN forgets information”. The diffusion term can be regarded as a mechanism for AntisymmetricRNNs to forget inputs in the past. As shown in the newly added Figure 2, when the diffusion constant increases, the eigenvalues of the end-to-end Jacobian decreases, resulting in shrinking gradient w.r.t. inputs in the past. In our current formulation, the diffusion factor is a constant across all the time steps and dimensions, but we could extend it to be time-dependent and/or data-dependent in future work.
>
> “better baseline in Cooijman et al., (2016)”? Thanks for the pointer. We added that in the footnote. We decide to keep the LSTM baseline reported by Arjovsky et al., (2016) because it has a higher accuracy on the more challenging pMNIST task than that in Cooijman et al., (2016) (92.6% vs 90.2%). We added the 92.6% accuracy in the footnote. Cooijman et al., (2016) is very relevant to our paper and we have added it to the related work section. It would be interesting to compare a “batch-normalized AntisymmetricRNN” with the batch-normalized LSTM in future work.

---

### Meta-Review · Area_Chair1 · 2018-12-17
**Accept**

**Confidence:** 5
**Recommendation:** Accept (Poster)

**Metareview:**

The paper presents a novel idea with a compelling experimental study. Good paper, accept.